# Provable Gaussian Embedding with One Observation

**Ming Yu** [*]    **Zhuoran Yang** [†]    **Tuo Zhao** [‡]    **Mladen Kolar** [§]    **Zhaoran Wang** [¶]

## Abstract

The success of machine learning methods heavily relies on having an appropriate representation for data at hand. Traditionally, machine learning approaches relied on user-defined heuristics to extract features encoding structural information about data. However, recently there has been a surge in approaches that learn how to encode the data automatically in a low dimensional space. Exponential family embedding provides a probabilistic framework for learning low-dimensional representation for various types of high-dimensional data [20]. Though successful in practice, theoretical underpinnings for exponential family embeddings have not been established. In this paper, we study the Gaussian embedding model and develop the first theoretical results for exponential family embedding models. First, we show that, under mild condition, the embedding structure can be learned from *one observation* by leveraging the parameter sharing between different contexts even though the data are *dependent* with each other. Second, we study properties of two algorithms used for learning the embedding structure and establish convergence results for each of them. The first algorithm is based on a convex relaxation, while the other solved the non-convex formulation of the problem directly. Experiments demonstrate the effectiveness of our approach.

## 1 Introduction

Exponential family embedding is a powerful technique for learning a low dimensional representation of high-dimensional data [20]. Exponential family embedding framework comprises of a *known* graph $G = (V, E)$ and the conditional exponential family. The graph $G$ has $m$ vertices and with each vertex we observe a $p$-dimensional vector $x_j$, $j = 1, \ldots, m$, representing an observation for which we would like to learn a low-dimensional embedding. The exponential family distribution is used to model the conditional distribution of $x_j$ given the context $\{x_k, (k, j) \in E\}$ specified by the neighborhood of the node $j$ in the graph $G$. In order for the learning of the embedding to be possible, one furthermore assumes how the parameters of the conditional distributions are shared across different nodes in the graph. The graph structure, conditional exponential family, and the way parameters are shared across the nodes are modeling choices and are application specific.

For example, in the context of word embeddings [1, 11], a word in a document corresponds to a node in a graph with the corresponding vector $x_j$ being a one-hot vector (the indicator of this word); the context of the word $j$ is given by the surrounding words and hence the neighbors of the node $j$ in the graph are the nodes corresponding to those words; and the conditional distribution of $x_j$ is

---

[*]Booth School of Business, University of Chicago, Chicago, IL. Email: `ming93@uchicago.edu`

[†]Department of Operations Research and Financial Engineering, Princeton University, Princeton, NJ.

[‡]School of Industrial and Systems Engineering, Georgia Institute of Technology, Atlanta, GA.

[§]Booth School of Business, University of Chicago, Chicago, IL.

[¶]Department of Industrial Engineering and Management Sciences, Northwestern University, Evanston, IL.

a multivariate categorical distribution. As another example arising in computational neuroscience consider embedding activities of neurons. Here the graph representing the context encodes spatial proximity of neurons and the Gaussian distribution is used to model the distributions of a neuron's activations given the activations of nearby neurons.

While exponential family embeddings have been successful in practice, theoretical underpinnings have been lacking. This paper is a step towards providing a rigorous understanding of exponential family embedding in the case of Gaussian embedding. We view the framework of exponential family embeddings through the lens of probabilistic graphical models [6], with the context graph specifying the conditional independencies between nodes and the conditional exponential family specifying the distribution locally. We make several contributions:

**1)** First, since the exponential family embedding specifies the distribution for each object conditionally on its context, there is no guarantee that there is a joint distribution that is consistent with all the conditional models. The probabilistic graphical models view allows us to provide conditions under which the conditional distributions defined a valid joint distribution over all the nodes.

**2)** Second, the probabilistic graphical model view allows us to learn the embedding vector from one observation — we get to see only one vector $x_j$ for each node $j \in V$ — by exploiting the shared parameter representation between different nodes of the graph. One might mistakenly then think that we in fact have $m$ observations to learn the embedding. However, the difficulty lies in the fact that these observations are not independent and the dependence intricately depends on the graph structure. Apparently not every graph structure can be learned from one observation, however, here we provide sufficient conditions on the graph that allow us to learn Gaussian embedding from one observation.

**3)** Finally, we develop two methods for learning the embedding. Our first algorithm is based on a convex optimization algorithm, while the second algorithm directly solves a non-convex optimization problem. They both provably recover the underlying embedding, but in practice, non-convex approach might lead to a faster algorithm.

## 1.1 Related Work

**Exponential family embedding**  Exponential family embedding originates from word embedding, where words or phrases from the vocabulary are mapped to embedding vectors [1]. Many variants and extensions of word embedding have been developed since [12, 9, 31, 10]. [20] develop a probabilistic framework based on general exponential families that is suitable for a variety of high-dimensional distributions, including Gaussian, Poisson, and Bernoulli embedding. This generalizes the embedding idea to a wider range of applications and types of data, such as real-valued data, count data, and binary data [13, 18, 19]. In this paper, we contribute to the literature by developing theoretical results on Gaussian embedding, which complements existing empirical results in the literature.

**Graphical model.**  The exponential family embedding is naturally related to the literature on probabilistic graphical models as the context structure forms a conditional dependence graph among the nodes. These two models are naturally related, but the goals and estimation procedures are very different. Much of the research effort on graphical model focus on learning the graph structure and hence the conditional dependency among nodes [8, 25, 29, 22]. As a contrast, in this paper, we instead focus on the problem where the graph structure is known and learn the embedding.

**Low rank matrix estimation.**  As will see in Section 2, the conditional distribution in exponential family embedding takes the form $f(VV^{\top})$ for the embedding parameter $V \in \mathbb{R}^{p \times r}$ which embeds the $p$ dimensional vector $x_j$ to $r$ dimensional space. Hence this is a low rank matrix estimation problem. Traditional methods focused on convex relaxation with nuclear norm regularization [14, 3, 17]. However, when the dimensionality is large, solving convex relaxation problem is usually time consuming. Recently there has been a lot of research on non-convex optimization formulations, from both theoretical and empirical perspectives [24, 26, 21, 27, 30]. People found that non-convex optimization is computationally more tractable, while giving comparable or better result. In our paper we consider both convex relaxation and non-convex optimization approaches.

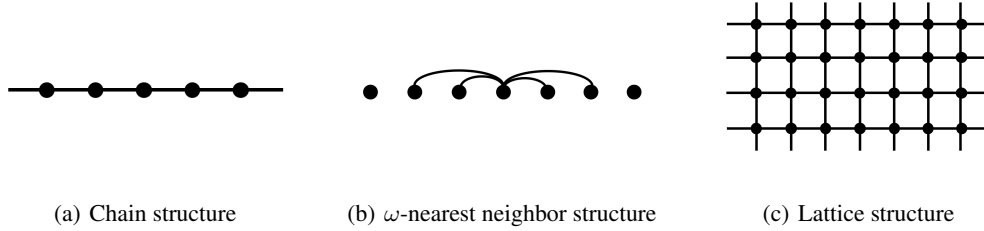

<div align="center">

(a) Chain structure       (b) $\omega$-nearest neighbor structure       (c) Lattice structure

Figure 1: Some commonly used context structures

</div>

## 2  Background

In this section, we briefly review the exponential family embedding framework. Let $X = (x_1, \ldots, x_m) \in \mathbb{R}^{p \times m}$ be the data matrix where a column $x_j \in \mathbb{R}^p$ corresponds to a vector observed at node $j$. For example, in word embedding, $x$ represents a document consisting of $m$ words, $x_j$ is a one-hot vector representation of the $j$-th word, and $p$ is the size of the dictionary. For each $j$, let $c_j \subseteq \{1, ..., m\}$ be the context of $j$, which is assumed to be known and is given by the graph $G$ — in particular, $c_j = \{k \in V : (j, k) \in E\}$. Some commonly used context structures are shown in Figure 1. Figure 1(a) is for chain structure. Note that this is different from vector autoregressive model where the chain structure is directed. Figure 1(b) is for $\omega$-nearest neighbor structure, where each node is connected with its preceding and subsequent $\omega$ nodes. This structure is common in word embedding where the preceding and subsequent $\omega$ words are the contexts. When $\omega = 1$ it boils down to the chain structure. Finally Figure 1(c) is for lattice structure that is widely used in the Ising model.

The exponential family embedding model assumes that $x_j$ conditioning on $x_{c_j}$ follows an exponential family distribution

$$x_j | x_{c_j} \sim \texttt{ExponentialFamily}\Big[\eta_j(x_{c_j}), t(x_j)\Big], \tag{2.1}$$

where $t(x_j)$ is the sufficient statistics and $\eta_j(x_{c_j}) \in \mathbb{R}^p$ is the natural parameter. For the linear embedding, we assume that $\eta$ in (2.1) takes the form

$$\eta_j(x_{c_j}) = f_j\Big(V_j \sum_{k \in c_j} V_k^\top x_k\Big), \tag{2.2}$$

where the link function $f_j$ is applied elementwise and $V_j \in \mathbb{R}^{p \times r}$. The low dimensional matrix $V_k$ embeds the vector $x_k \in \mathbb{R}^p$ to a lower $r$-dimensional space with $V_k^\top x_k \in \mathbb{R}^r$ being the embedding of $x_k$. For example, in word embedding each row of $V_k$ is the embedding rule for a word. Since $x_k$ is a one-hot vector, we see that $V_k^\top x_k$ is selecting a row of $V_k$ that corresponds to the word on the node $k$. A common simplifying assumption is that the embedding structure is shared across the nodes by assuming that $V_j = V$ for all $j \in V$. In word-embedding, this makes the embedding rule not depend on the position of the word in the document. We summarize some commonly seen exponential family distributions and show how they define an exponential family embedding model.

**Gaussian embedding.**  In Gaussian embedding it is assumed that the conditional distribution is

$$x_j | x_{c_j} \sim N\Big(V \sum_{k \in c_j} V^\top x_k, \Sigma_j\Big) = N\Big(M \sum_{k \in c_j} x_k, \Sigma_j\Big), \tag{2.3}$$

where $M = VV^\top$ and $\Sigma_j$ is the conditional covariance matrix for each node $j$. We will prove in Section 3 that under mild conditions, these conditional distributions define a valid joint Gaussian distribution. The link function for Gaussian embedding is the identity function, but one may choose the link function to be $f(\cdot) = \log(\cdot)$ in order to constrain the parameters to be non-negative. Gaussian embedding is commonly applied to real valued observations.

**Word embedding (cbow [11]).** In the word embedding setting, $x_j$ is an indicator of the $j$-th word in a document and the dimension of $x_j$ is equal to the size of the vocabulary. The context of the $j$-th word, $c_j$, is the window of size $\omega$ around $x_j$, that is, $c_j = \{k \in \{1, ..., m\}\colon k \neq j, |k - j| \leq \omega\}$. Cbow is a special case of exponential family embedding with

$$p(x_j|x_{c_j}) = \frac{\exp\left[x_j^\top V \left(\sum_{k \in c_j} V^\top x_k\right)\right]}{\sum_j \exp\left[x_j^\top V \left(\sum_{k \in c_j} V^\top x_k\right)\right]}. \tag{2.4}$$

**Poisson embedding.** In Poisson embedding, the sufficient statistic is the identity and the natural parameter is the logarithm of the rate. The conditional distribution is given as

$$x_j|x_{c_j} \sim \text{Poisson}\Big(\exp\big(V \sum_{k \in c_j} V^\top x_k\big)\Big). \tag{2.5}$$

Poisson embedding can be applied to count data.

## 3 Gaussian Embedding Model

In this paper, we consider the case of Gaussian embedding, where the conditional distribution of $x_j$ given its context $x_{c_j}$ is given in (2.3) with the conditional covariance matrix $\Sigma_j$ unknown. The parameter matrix $M = VV^\top$ with $V \in \mathbb{R}^{p \times r}$ will be learned from the data matrix $X \in \mathbb{R}^{p \times m}$ and $V^\top x_k$ is the embedding of $x_k$.

Let $X_{\text{col}} = [x_1^\top, x_2^\top, ..., x_m^\top]^\top \in \mathbb{R}^{pm \times 1}$ be the column vector obtained by stacking columns of $X$ and let $[x_j]_\ell$ denote the $\ell$-th coordinate of $x_j$. We first restate a definition on compatibility from [23].

**Definition 3.1.** A non-negative function $g$ is capable of generating a conditional density function $p(y|x)$ if

$$p(y|x) = \frac{g(y, x)}{\int g(y, x) dy}. \tag{3.1}$$

Two conditional densities are said to be compatible if there exists a function $g$ that can generate both conditional densities. When $g$ is a density, the conditional densities are called strongly compatible.

Since $M$ is symmetric, according to Proposition 1 in [4], we have the following proposition.

**Proposition 3.2.** The conditional distributions (2.3) is compatible and the joint distribution of $X_{\text{col}}$ is of the form $p(x_{\text{col}}) \propto \exp\left(-\frac{1}{2}x_{\text{col}}^\top \cdot \Sigma_{\text{col}}^{-1} \cdot x_{\text{col}}\right)$ for some $\Sigma_{\text{col}} \in \mathbb{R}^{pm \times pm}$. When the choice of $M$ and $\Sigma_j$ is such that $\Sigma_{\text{col}} \succ 0$, the conditional distributions are strongly compatible and we have $X_{\text{col}} \sim N(0, \Sigma_{\text{col}})$.

The explicit expression of $\Sigma_{\text{col}}$ can be derived from (2.3), however, in general is quite complicated. The following example provides an explicit formula in the case where $\Sigma_j = I$.

**Example 3.3.** Suppose that $\Sigma_j = I$ for all $j = 1, \ldots, m$. Let $A \in \mathbb{R}^{m \times m}$ denote the adjacency matrix of the graph $G$, with $a_{j,k} = 1$ when there is an edge between nodes $j$ and $k$ and $0$ otherwise. Denote $\ell^c = \{1, \ldots, \ell - 1, \ell + 1, \ldots, p\}$, the conditional distribution of $[x_j]_\ell$ is given by

$$[x_j]_\ell \,\Big|\, x_{c_j}, [x_j]_{\ell^c} \sim N\left(\Big[M \sum_{k \in c_j} x_k\Big]_\ell, 1\right).$$

Moreover, there exists a joint distribution $X_{\text{col}} \sim N(0, \Sigma_{\text{col}})$ where $\Sigma_{\text{col}} \in \mathbb{R}^{pm \times pm}$ satisfies

$$\Sigma_{\text{col}}^{-1} = I - A \otimes M, \tag{3.2}$$

and $A \otimes M$ denotes the Kronecker product between $A$ and $M$. Clearly, we need $\Sigma_{\text{col}} \succ 0$, which imposes implicit restrictions on $A$ and $M$. To ensure that $\Sigma_{\text{col}}$ is positive definite, we need to

ensure that all the eigenvalues of $A \otimes M$ are smaller than 1. One sufficient condition for this is $\|A\|_2 \cdot \|M\|_2 < 1$. For example, consider a chain graph with

$$A = \begin{bmatrix} 0 & 1 & & \\ 1 & 0 & \ddots & \\ & \ddots & \ddots & 1 \\ & & 1 & 0 \end{bmatrix} \in \mathbb{R}^{p \times p} \text{ and } \Sigma_{\text{col}}^{-1} = \begin{bmatrix} I & -M & & \\ -M & I & \ddots & \\ & \ddots & \ddots & -M \\ & & -M & I \end{bmatrix} \in \mathbb{R}^{pm \times pm}. \quad (3.3)$$

Then it suffices to have $\|M\|_2 < 1/2$. Similarly for $\omega$-nearest neighbor structure, it suffices to have $\|M\|_2 < 1/2\omega$ and for the lattice structure to have $\|M\|_2 < 1/4$.

## 3.1 Estimation Procedures

Since $\Sigma_j$ is unknown, we propose to minimize the following loss function based on the conditional log-likelihood

$$\mathcal{L}(M) = m^{-1} \sum_{j=1}^{m} \mathcal{L}^j(M), \quad (3.4)$$

where $\mathcal{L}^j(M) := \frac{1}{2} \cdot \left\| x_j - M \sum_{k \in c_j} x_k \right\|^2$. Let $M^* = V^* V^{*\top}$ denote the true rank $r$ matrix with $V^* \in \mathbb{R}^{p \times r}$. Note that $V^*$ is not unique, but $M^*$ is. Observe that minimizing (3.4) leads to a consistent estimator, since

$$\mathbb{E}\left[ \nabla \mathcal{L}^j(M^*) \right] = \mathbb{E}\left[ \left( x_j - M^* \sum_{k \in c_j} x_k \right) \sum_{k \in c_j} x_k^\top \right] = \mathbb{E}_{x_{c_j}} \mathbb{E}_{x_j} \left[ \left( x_j - M^* \sum_{k \in c_j} x_k \right) \sum_{k \in c_j} x_k^\top \,\Big|\, x_{c_j} \right] = 0.$$

In order to find a low rank solution $\widehat{M}$ that approximates $M^*$, we consider the following two procedures.

**Convex Relaxation**    We solve the following problem

$$\min_{M \in \mathbb{R}^{p \times p}, M^\top = M, M \succeq 0} \mathcal{L}(M) + \lambda \|M\|_*, \quad (3.5)$$

where $\| \cdot \|_*$ is the nuclear norm of a matrix and $\lambda$ is the regularization parameter to be specified in the next section. The problem (3.5) is convex and hence can be solved by proximal gradient descent method [15] with any initialization point.

**Non-convex Optimization**    Although it is guaranteed to find global minimum by solving the convex relaxation problem (3.5), in practice it may be slow. In our problem, since $M$ is low rank and positive semidefinite, we can always write $M = VV^\top$ for some $V \in \mathbb{R}^{p \times r}$ and solve the following non-convex problem

$$\min_{V \in \mathbb{R}^{p \times r}} \mathcal{L}(VV^\top). \quad (3.6)$$

With an appropriate initialization $V^{(0)}$, in each iteration we update $V$ by gradient descent

$$V^{(t+1)} = V^{(t)} - \eta \cdot \nabla_V \mathcal{L}(VV^\top)\big|_{V=V^{(t)}},$$

where $\eta$ is the step size. The choice of initialization $V^{(0)}$ and step size $\eta$ will be specified in details in the next section. The unknown rank $r$ can be estimated as in [2].

## 4    Theoretical Result

We establish convergence rates for the two estimation procedures.

## 4.1 Convex Relaxation

In order to show that a minimizer of (3.5) gives a good estimator for $M$, we first show that the objective function $\mathcal{L}(\cdot)$ is strongly convex under the assumption that the data are distributed according to (2.3) with the true parameter $M^* = V^* V^{*\top}$ with $V^* \in \mathbb{R}^{p \times r}$. Let

$$\delta\mathcal{L}(\Delta) = \mathcal{L}(M^* + \Delta) - \mathcal{L}(M^*) - \langle \nabla\mathcal{L}(M^*), \Delta \rangle,$$

where $\langle A, B \rangle = \text{tr}(A^\top B)$ and $\Delta$ is a symmetric matrix. Let $\Delta_i$ denote the $i$-th column of $\Delta$ and let $\Delta_{\text{col}} = [\Delta_1^\top, \ldots, \Delta_p^\top]^\top \in \mathbb{R}^{p^2 \times 1}$ be the vector obtained by stacking columns of $\Delta$. Then a simple calculation shows that

$$\delta\mathcal{L}(\Delta) = \frac{1}{m} \cdot \sum_{j=1}^{m} \left\| \Delta \sum_{k \in c_j} x_k \right\|^2 = \sum_{i=1}^{p} \Delta_i^\top \left[ \frac{1}{m} \sum_{j=1}^{m} \left( \sum_{k \in c_j} x_k \right) \cdot \left( \sum_{k \in c_j} x_k \right)^\top \right] \Delta_i$$

has a quadratic form in each $\Delta_i$ with the same Hessian matrix $H$. Let

$$\widetilde{X} = \left[ \sum_{k \in c_1} x_k, \sum_{k \in c_2} x_k, \ldots, \sum_{k \in c_m} x_k \right] = X \cdot A \in \mathbb{R}^{p \times m},$$

where $A$ is the adjacency matrix of the graph $G$. Then the Hessian matrix is given by

$$H = \frac{1}{m} \sum_{j=1}^{m} \left( \sum_{k \in c_j} x_k \right) \cdot \left( \sum_{k \in c_j} x_k \right)^\top = \frac{1}{m} \widetilde{X} \widetilde{X}^\top = \frac{1}{m} X A A^\top X^\top \in \mathbb{R}^{p \times p} \tag{4.1}$$

and therefore we can succinctly write $\delta\mathcal{L}(\Delta) = \Delta_{\text{col}}^\top \cdot H_{\text{col}} \cdot \Delta_{\text{col}}$, where the total Hessian matrix $H_{\text{col}} = \text{diag}(H, H, \ldots, H) \in \mathbb{R}^{p^2 \times p^2}$ is a block diagonal matrix.

To show that $\mathcal{L}(\cdot)$ is strongly convex, it suffices to lower bound the minimum eigenvalue of $H$, defined in (4.1). If the columns of $\widetilde{X}$ were independent, the minimum eigenvalue of $H$ would be bounded away from zero with overwhelming probability for a large enough $m$ [16]. However, in our setting the columns of $\widetilde{X}$ are dependent and we need to prove this lower bound using different tools. As the distribution of $X$ depends on the unknown conditional covariance matrices $\Sigma_j, j = 1, \ldots, m$ in a complicated way, we impose the following assumption on the expected version of $H$.

**Assumption EC.** The minimum and maximum eigenvalues of $\mathbb{E}H$ are bounded from below and from above: $0 < c_{\min} \le \sigma_{\min}(\mathbb{E}H) \le \sigma_{\max}(\mathbb{E}H) \le c_{\max} < \infty$.

Assumption (EC) puts restrictions on conditional covariance matrices $\Sigma_j$ and can be verified in specific instances of the problem. In the context of Example 3.3, where $\Sigma_j = I, j = 1, \ldots, m$, and the graph is a chain, we have the adjacency matrix $A$ and the covariance matrix $\Sigma_{\text{col}}$ given in (3.3). Then simple linear algebra [5] gives us that

$$\mathbb{E}H = m^{-1} \mathbb{E} X A A^\top X^\top = 2I + cM^2 + o(M^2),$$

which guarantees that $\sigma_{\min}(\mathbb{E}H) \ge 1$ and $\sigma_{\max}(\mathbb{E}H) \le c + 3$ for large enough $m$.

The following assumption requires that the spectral norm of $A$ and $\Sigma_{\text{col}}$ do not scale with $m$.

**Assumption SC.** There exists a constant $\rho_0$ such that $\max\left\{ \|A\|_2, \|\Sigma_{\text{col}}^{1/2}\|_2 \right\} \le \rho_0$.

Assumption (SC) gives sufficient condition on the graph structure, and it requires that the dependency among nodes is weak. In fact, it can be relaxed to $\rho_0 = o(m^{1/4})$ which allows the spectral norm to scale with $m$ slowly. In this way, the minimum and maximum eigenvalues in assumption (EC) also scale with $m$ and it results in a much larger sample complexity on $m$. However, if $\rho_0$ grows even faster, then there is no way to guarantee a reasonable estimation. We see that $\rho_0 \sim m^{1/4}$ is the critical condition, and we have the phase transition on this boundary.

It is useful to point out that these assumptions are not restrictive. For example, under the simplification that $\Sigma_j = I$, we have $\|\Sigma_{\text{col}}\|_2 = 1/(1 - \|A\|_2 \cdot \|M\|_2)$. The condition $\|A\|_2 \cdot \|M\|_2 < 1$ is satisfied

naturally for a valid Gaussian embedding model. Therefore in order to have $\|\Sigma_{\text{col}}^{1/2}\|_2 \leq \rho_0$, we only need that $\|A\|_2 \cdot \|M\|_2 \leq 1 - 1/\rho_0^2$, i.e., it is bounded away from 1 by a constant distance.

It is straightforward to verify that assumption (SC) holds for the chain structure in Example 3.3. If the graph is fully connected, we have $\|A\|_2 = m - 1$, which violates the assumption. In general, assumption (SC) gives a sufficient condition on the graph structure so that the embedding is learnable.

With these assumptions, the following lemma proves that the minimum and maximum eigenvalues of the sample Hessian matrix $H$ are also bounded from below and above with high probability.

**Lemma 4.1.** Suppose the assumption (EC) and (SC) hold. Then for $m \geq c_0 p$ we have $\sigma_{\min}(H) \geq \frac{1}{2}c_{\min}$ and $\sigma_{\max}(H) \leq 2c_{\max}$ with probability at least $1 - c_1 \exp(-c_2 m)$, where $c_0, c_1, c_2$ are absolute constants. Therefore

$$\kappa_\mu \cdot \|\Delta\|_F^2 \leq \delta\mathcal{L}(\Delta) \leq \kappa_L \cdot \|\Delta\|_F^2, \tag{4.2}$$

with $\kappa_\mu = \frac{1}{2}c_{\min}$ and $\kappa_L = 2c_{\max}$ for any $\Delta \in \mathbb{R}^{p \times p}$.

Lemma 4.1 is the key technical result, which shows that although all the $x_j$ are dependent, the objective function $\mathcal{L}(\cdot)$ is still strongly convex and smooth in $\Delta$. Since the loss function $\mathcal{L}(\cdot)$ is strongly convex, an application of Theorem 1 in [14] gives the following result on the performance of the convex relaxation approach proposed in the previous section.

**Theorem 4.2.** Suppose the assumptions (SC) and (EC) are satisfied. The minimizer $\widehat{M}$ of (3.5) with $\lambda \geq \left\| \frac{1}{m} \sum_{j=1}^m \left( x_j - M^* \sum_{k \in c_j} x_k \right) \cdot \sum_{k \in c_j} x_k^\top \right\|_2$ satisfies

$$\|\widehat{M} - M^*\|_F \leq \frac{32\sqrt{r}\lambda}{\kappa_\mu}.$$

The following lemma gives us a way to set the regularization parameter $\lambda$.

**Lemma 4.3.** Let $G = \frac{1}{m} \sum_{j=1}^m \Sigma_j$. Assume that the maximum eigenvalue of $G$ is bounded from above as $\sigma_{\max}(G) \leq \eta_{\max}$ for some constant $\eta_{\max}$. Then there exist constants $c_0, c_1, c_2, c_3 > 0$ such that for $m \geq c_0 p$, we have

$$\mathbb{P}\left[ \left\| \frac{1}{m} \sum_{j=1}^m \left( x_j - M^* \sum_{k \in c_j} x_k \right) \cdot \sum_{k \in c_j} x_k^\top \right\|_2 \geq c_1 \sqrt{\frac{p}{m}} \right] \leq c_2 \exp(-c_3 m).$$

Combining the result of Lemma 4.3 with Theorem 4.2, we see that $\lambda$ should be chosen as $\lambda = \mathcal{O}\left(\sqrt{p/m}\right)$, which leads to the error rate

$$\|\widehat{M} - M^*\|_F = \mathcal{O}_P\left( \frac{1}{\kappa_\mu} \sqrt{\frac{rp}{m}} \right). \tag{4.3}$$

## 4.2 Non-convex Optimization

Next, we consider the convergence rate for the non-convex method resulting in minimizing (3.6) in $V$. Since the factorization of $M^*$ is not unique, we measure the subspace distance between $V$ and $V^*$.

**Subspace distance.** Let $V^*$ be such that $V^* V^{*\top} = \Theta^*$. Define the subspace distance between $V$ and $V^*$ as

$$d^2(V, V^*) = \min_{O \in \mathcal{O}(r)} \|V - V^* O\|_F^2, \tag{4.4}$$

where $\mathcal{O}(r) = \{O : O \in \mathbb{R}^{r \times r}, OO^\top = O^\top O = I\}$.

Next, we introduce the notion of the statistical error. Denote

$$\Omega = \left\{ \Delta : \Delta \in \mathbb{R}^{p \times p}, \Delta = \Delta^\top, \text{rank}(\Delta) = 2r, \|\Delta\|_F = 1 \right\}.$$

The statistical error is defined as

$$e_{\text{stat}} = \sup_{\Delta \in \Omega} \left\langle \nabla\mathcal{L}(M^*), \Delta \right\rangle. \tag{4.5}$$

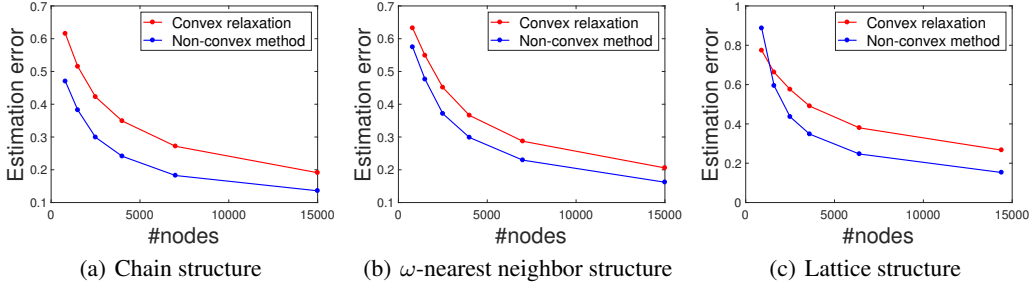

(a) Chain structure      (b) $\omega$-nearest neighbor structure      (c) Lattice structure

Figure 2: Estimation accuracy for three context structures

Intuitively, the statistical error quantifies how close the estimator can be to the true value. Specifically, if $V$ is within $c \cdot e_{\text{stat}}$ distance from $V^*$, then it is already optimal. For any $\Delta \in \Omega$, we have the factorization $\Delta = U_\Delta V_\Delta^\top$ where $U_\Delta, V_\Delta \in \mathbb{R}^{p \times 2r}$ and $\|U_\Delta\|_2 = \|V_\Delta\|_F = 1$. We then have

$$
\begin{aligned}
\langle \nabla \mathcal{L}(M^*), \Delta \rangle = \langle \nabla \mathcal{L}(M^*) V_\Delta, U_\Delta \rangle &\leq \|\nabla \mathcal{L}(M^*) V_\Delta\|_F \cdot \|U_\Delta\|_F \\
&\leq \|\nabla \mathcal{L}(M^*)\|_2 \|V_\Delta\|_F \|U_\Delta\|_F \leq \sqrt{2r}\lambda,
\end{aligned}
\tag{4.6}
$$

where the last inequality follows from Lemma 4.3. In particular, we see that both convex relaxation and non-convex optimization give the same rate.

**Initialization.** In order to prove a linear rate of convergence for the procedure, we need to initialize it properly. Since the loss function $\mathcal{L}(M)$ is quadratic in $M$, we can ignore all the constraints on $M$ and get a closed form solution as

$$
M^{(0)} = \Big[ \sum_{j=1}^m \Big( \sum_{k \in c_j} x_k \Big) \Big( \sum_{k \in c_j} x_k \Big)^\top \Big]^{-1} \cdot \Big[ \sum_{j=1}^m x_j^\top \Big( \sum_{k \in c_j} x_k \Big) \Big].
\tag{4.7}
$$

We then apply rank-$r$ eigenvalue decomposition on $\widetilde{M}^{(0)} = \frac{1}{2}\big(M^{(0)} + M^{(0)\top}\big)$ and obtain $[\widetilde{V}, \widetilde{S}, \widetilde{V}] = \text{rank-}r$ svd of $\widetilde{M}^{(0)}$. Then $V^{(0)} = \widetilde{V}\widetilde{S}^{1/2}$ is the initial point for the gradient descent. The following lemma quantifies the accuracy of this initialization.

**Lemma 4.4.** The initialization $M^{(0)}$ and $V^{(0)}$ satisfy

$$
\|M^{(0)} - M^*\|_F \leq \frac{2\sqrt{p}\lambda}{\kappa_\mu} \quad \text{and} \quad d^2\big(V^{(0)}, V^*\big) \leq \frac{20p\lambda^2}{\kappa_\mu^2 \cdot \sigma_r(M^*)}
$$

where $\sigma_r(M^*)$ is the minimum non-zero singular value of $M^*$.

With this initialization, we obtain the following main result for the non-convex optimization approach, which establishes a linear rate of convergence to a point that has the same statistical error rate as the convex relaxation approach studied in Theorem 4.2.

**Theorem 4.5.** Suppose the assumption (EC) and (SC) are satisfied, and suppose the step size $\eta$ satisfies $\eta \leq \big[32\|M^{(0)}\|_2^2 \cdot (\kappa_\mu + \kappa_L)\big]^{-1}$. For large enough $m$, after $T$ iterations we have

$$
d^2\big(V^{(T)}, V^*\big) \leq \beta^T d^2\big(V^{(0)}, V^*\big) + \frac{C}{\kappa_\mu^2} \cdot e_{\text{stat}}^2,
\tag{4.8}
$$

for some constant $\beta < 1$ and a constant $C$.

# 5 Experiment

In this section, we evaluate our methods through experiments. We first justify that although $\Sigma_j$ is unknown, minimizing (3.4) still leads to a consistent estimator. We compare the estimation accuracy

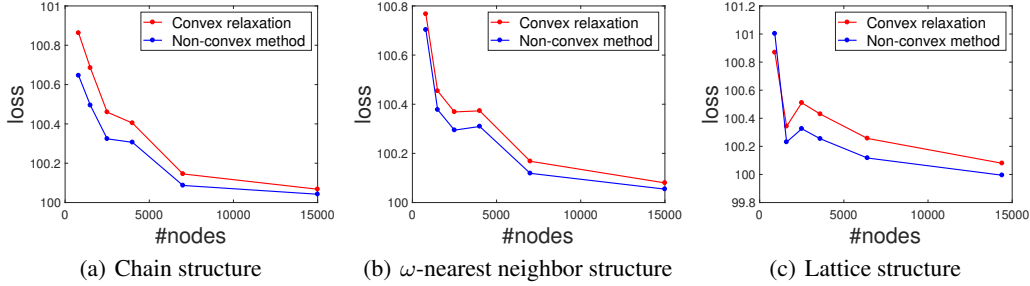

|                        |                                |                      |
|:----------------------:|:------------------------------:|:--------------------:|
| (a) Chain structure    | (b) $\omega$-nearest neighbor structure | (c) Lattice structure |

Figure 3: Testing loss for three context structures

with known and unknown covariance matrix $\Sigma_j$. We set $\Sigma_j = \sigma_j \cdot \text{Toeplitz}(\rho_j)$ where $\text{Toeplitz}(\rho_j)$ denotes Toeplitz matrix with parameter $\rho_j$. We set $\rho_j \sim U[0, 0.3]$ and $\sigma_j \sim U[0.4, 1.6]$ to make them non-isotropic. The estimation accuracy with known and unknown $\Sigma_j$ are given in Table 1. We can see that although knowing $\Sigma_j$ could give slightly better accuracy, the difference is tiny. Therefore, even if the covariance matrices are not isotropic, ignoring them still gives a consistent estimator.

Table 1: Comparison of estimation accuracy with known and unknown covariance matrix

|          | $m = 1000$ | $m = 2500$ | $m = 5000$ | $m = 8000$ | $m = 15000$ |
|---------:|:----------:|:----------:|:----------:|:----------:|:-----------:|
| unknown  | 0.8184     | 0.4432     | 0.3210     | 0.2472     | 0.1723      |
| known    | 0.7142     | 0.3990     | 0.2908     | 0.2288     | 0.1649      |

We then consider three kinds of graph structures given in Figure 1: chain structure, $\omega$-nearest neighbor structure, and lattice structure. We generate the data according to the conditional distribution (2.3) using Gibbs Sampling. We set $p = 100, r = 5$ and vary the number of nodes $m$. For each $j$, we set $\Sigma_j = \Sigma$ to be a Toeplitz matrix with $\Sigma_{i\ell} = \rho^{|i-\ell|}$ with $\rho = 0.3$. We generate independent train, validation, and test sets. For convex relaxation, the regularization parameter is selected using the validation set. We consider two metrics, one is the estimation accuracy $\|\widehat{M} - M^*\|_F / \|M^*\|_F$, and the other is the loss $\mathcal{L}(\widehat{M})$ on the test set.

The simulation results for estimation accuracy for the three graph structures are shown in Figure 2, and the results for loss on test sets are shown in Figure 3. Each result is based on 20 replicates. For the estimation accuracy, we see that when the number of nodes is small, neither method gives accurate estimation; for reasonably large $m$, non-convex method gives better estimation accuracy since it does not introduce bias; for large enough $m$, both methods give accurate and similar estimation. For the loss on test sets, we see that in general, both methods give smaller loss as $m$ increases. The non-convex method gives marginally better loss. This demonstrates the effectiveness of our methods.

## 6  Conclusion

In this paper, we focus on Gaussian embedding and develop the first theoretical result for exponential family embedding model. We show that for various kinds of context structures, we are able to learn the embedding structure with only one observation. Although all the data we observe are dependent, we show that the objective function is still well-behaved and therefore we can learn the embedding structure reasonably well.

It is useful to point out that, the theoretical framework we proposed is for *general exponential family embedding models*. As long as the similar conditions are satisfied, the framework and theoretical results hold for any general exponential family embedding model as well. However, proving these conditions is quite challenging from the probability perspective. Nevertheless, our framework still holds and all we need are more complicated probability tools. Extending the result to other embedding models, for example the Ising model, is work in progress.

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
