[Supplementary Material · GaussianEmbedding_supplement.pdf]

## A  Technical proofs

### A.1  Proof of Lemma 4.1.

*Proof.* Before we proceed with the main proof, we first introduce the following lemma in [7].

**Lemma A.1.** Let $x_1, ..., x_T$ be independent and identically drawn from distribution $N(0,1)$ and $X = (x_1, ..., x_T)^\top$ be a random vector. Suppose a function $f : \mathbb{R}^T \to \mathbb{R}$ is Lipschitz, i.e., for any $v_1, v_2 \in \mathbb{R}^T$, there exists $L$ such that $|f(v_1) - f(v_2)| \le L\|v_1 - v_2\|_2$, then we have that

$$\mathbb{P}\Big\{|f(X) - \mathbb{E}f(X)| > t\Big\} \le 2\exp\Big(-\frac{t^2}{2L^2}\Big)$$

for all $t > 0$.

We then proceed with the proof of Lemma 4.1. For any fixed $v \in \mathbb{R}^p$ with $\|v\|_2 = 1$, define

$$W = f_v(Z) = \frac{1}{\sqrt{m}}\Big\|v^\top \operatorname{mat}\big(\Sigma_{\text{col}}^{1/2} Z\big) \cdot A\Big\|_2,$$

where $Z \in \mathbb{R}^{pm \times 1}$ and $\operatorname{mat}(\cdot)$ is a reshape operator that reshape a $pm$-dimensional vector to a $p \times m$ dimensional matrix. When $Z \sim N(0, I_{pm})$, it is straightforward to see that the distribution of $\operatorname{mat}\big(\Sigma_{\text{col}}^{1/2} Z\big)$ is the same as $X$ and hence $W^2$ has the same distribution with $v^\top H v$. We then verify that the function $f_v$ is Lipschitz with $L = \frac{\rho_0^2}{\sqrt{m}}$ where $\rho_0$ is defined in assumption (SC). For any vector $Z_1, Z_2$, we have

$$\begin{aligned}
\Big|f_v(Z_1) - f_v(Z_2)\Big| &= \frac{1}{\sqrt{m}}\Big|\big\|v^\top re\big(\Sigma_{\text{col}}^{1/2} Z_1\big) \cdot A\big\|_2 - \big\|v^\top re\big(\Sigma_{\text{col}}^{1/2} Z_2\big) \cdot A\big\|_2\Big| \\
&\le \frac{1}{\sqrt{m}}\Big|v^\top re\big(\Sigma_{\text{col}}^{1/2}(Z_1 - Z_2)\big) \cdot A\Big| \\
&\le \frac{1}{\sqrt{m}}\|v\|_2\Big\|\Sigma_{\text{col}}^{1/2}(Z_1 - Z_2)\Big\|_2 \cdot \|A\|_2 \\
&\le \frac{1}{\sqrt{m}}\|\Sigma_{\text{col}}^{1/2}\|_2\|Z_1 - Z_2\|_2 \cdot \|A\|_2 \\
&= \frac{\rho_0^2}{\sqrt{m}}\|Z_1 - Z_2\|_2.
\end{aligned} \tag{A.1}$$

Using Lemma A.1, we have that

$$\mathbb{P}\Big\{|W - \mathbb{E}W| > t\Big\} \le 2\exp\Big(-\frac{t^2 m}{2\rho_0^4}\Big). \tag{A.2}$$

Since $W \ge 0$ and hence $\mathbb{E}W \ge 0$, we have

$$\Big[(\mathbb{E}W^2)^{1/2} - \mathbb{E}W\Big]^2 \le \Big[(\mathbb{E}W^2)^{1/2} + \mathbb{E}W\Big] \cdot \Big[(\mathbb{E}W^2)^{1/2} - \mathbb{E}W\Big] = \operatorname{Var}(W).$$

Moreover, from (A.2) we have

$$\operatorname{Var}(W) = \mathbb{E}\Big\{\big(W - \mathbb{E}W\big)^2\Big\} = \int_0^\infty \mathbb{P}\Big\{\big(W - \mathbb{E}W\big)^2 \ge t^2\Big\}d(t^2) \le \int_0^\infty 2\exp\Big(-\frac{t^2 m}{2\rho_0^4}\Big)d(t^2) = \frac{4\rho_0^4}{m},$$

and hence

$$(\mathbb{E}W^2)^{1/2} - \mathbb{E}W \le \frac{2\rho_0^2}{\sqrt{m}}. \tag{A.3}$$

According to (A.3), we know that $|W - \mathbb{E}W| \le t$ implies $|W - (\mathbb{E}W^2)^{1/2}| \le t + 2\rho_0^2/\sqrt{m}$, which gives

$$\mathbb{P}\Big(|W - (\mathbb{E}W^2)^{1/2}| > t + 2\rho_0^2/\sqrt{m}\Big) \le \mathbb{P}\Big(|W - \mathbb{E}W| > t\Big) \le 2\exp\Big(-\frac{t^2 m}{2\rho_0^4}\Big) \tag{A.4}$$

for any fixed $v \in \mathbb{R}^p$ with $\|v\|_2 = 1$. For large enough $m$, taking $t = \frac{1}{4}c_{\min}$ and apply union bound on 1/4-covering of $\mathbb{S}^{m-1} = \{v \in \mathbb{R}^m \mid \|v\|_2 = 1\}$ we completes the proof. The proof for upper bound is similar. □

## A.2 Proof of Lemma 4.3.

*Proof.* Before we proceed with the main proof, we first introduce the following lemma in [14].

**Lemma A.2** (Lemma I.2 in [14]). Given a Gaussian random vector $Y \sim N(0, S)$ with $Y \in \mathbb{R}^{m \times 1}$, for all $t > 2/\sqrt{m}$ we have

$$\mathbb{P}\left[\frac{1}{m}\left|\|Y\|_2^2 - \operatorname{tr} S\right| > 4t\|S\|_2\right] \leq 2\exp\left(-\frac{m\left(t - \frac{2}{\sqrt{m}}\right)^2}{2}\right) + 2\exp\left(-\frac{m}{2}\right). \tag{A.5}$$

We then proceed with the proof of Lemma 4.3. Denote $q_j = x_j - M^* \sum_{k \in c_j} x_k \sim N(0, \Sigma_j)$ and denote $Q = [q_1, ..., q_m] \in \mathbb{R}^{p \times m}$, we have $\mathbb{E}\frac{1}{m}QQ^\top = G$ and

$$\frac{1}{m}\sum_{j=1}^m \left(x_j - M^* \sum_{k \in c_j} x_k\right) \cdot \sum_{k \in c_j} x_k^\top = \frac{1}{m}Q \cdot \widetilde{X}. \tag{A.6}$$

For any fixed $v \in \mathbb{R}^p$ with $\|v\|_2 = 1$, we have

$$\frac{1}{m}v^\top Q\widetilde{X}v = \frac{1}{m}\sum_{j=1}^m v^\top q_j \cdot \widetilde{x}_j^\top v = \frac{1}{2m}\left[\sum_{j=1}^m \langle v, q_j + \widetilde{x}_j\rangle^2 - \sum_{j=1}^m \langle v, q_j\rangle^2 - \sum_{j=1}^m \langle v, \widetilde{x}_j\rangle^2\right]$$

$$= \underbrace{\frac{1}{2}v^\top\left(\frac{1}{m}\sum_{j=1}^m (q_j + \widetilde{x}_j)(q_j + \widetilde{x}_j)^\top\right)v - \frac{1}{2}v^\top\mathbb{E}(H + QQ^\top)v}_{R_1}$$

$$- \underbrace{\left[\frac{1}{2}v^\top\left(\frac{1}{m}\sum_{j=1}^m q_j q_j^\top\right)v - \frac{1}{2}v^\top\mathbb{E}QQ^\top \cdot v\right]}_{R_2} - \underbrace{\left[\frac{1}{2}v^\top\left(\frac{1}{m}\sum_{j=1}^m \widetilde{x}_j\widetilde{x}_j^\top\right)v - \frac{1}{2}v^\top\mathbb{E}H \cdot v\right]}_{R_3}$$

$$= R_1 - R_2 - R_3. \tag{A.7}$$

Each $R_j$ for $j = 1, 2, 3$ is a deviation term and can be bounded similarly. For $R_3$, define the random vector $Y \in \mathbb{R}^m$ with component $Y_j = v^\top \widetilde{x}_j$. Using Lemma A.2 and together with assumption EC, we obtain

$$\mathbb{P}\left[|R_3| > 4t\sigma_{\max}\right] \leq 2\exp\left(-\frac{m\left(t - \frac{2}{\sqrt{m}}\right)^2}{2}\right) + 2\exp\left(-\frac{m}{2}\right). \tag{A.8}$$

Similarly, for $R_1$ and $R_2$ we have

$$\mathbb{P}\left[|R_2| > 4t\eta_{\max}\right] \leq 2\exp\left(-\frac{m\left(t - \frac{2}{\sqrt{m}}\right)^2}{2}\right) + 2\exp\left(-\frac{m}{2}\right), \tag{A.9}$$

and

$$\mathbb{P}\left[|R_1| > 4t(\sigma_{\max} + \eta_{\max})\right] \leq 2\exp\left(-\frac{m\left(t - \frac{2}{\sqrt{m}}\right)^2}{2}\right) + 2\exp\left(-\frac{m}{2}\right). \tag{A.10}$$

Combine these three bounds, for fixed $v \in \mathbb{R}^p$ with $\|v\|_2 = 1$, we have

$$\mathbb{P}\left[\frac{1}{m}\left|v^\top Q\widetilde{X}v\right| > 8t(\sigma_{\max} + \eta_{\max})\right] \leq 6\exp\left(-\frac{m\left(t - \frac{2}{\sqrt{m}}\right)^2}{2}\right) + 6\exp\left(-\frac{m}{2}\right). \tag{A.11}$$

Setting $t = 4\sqrt{p/m}$ and taking the union bound on 1/4-covering of $\mathbb{S}^{m-1} = \{v \in \mathbb{R}^m \mid \|v\|_2 = 1\}$ completes the proof. □

## A.3 Proof of Lemma 4.4.

*Proof.* Since $M^{(0)}$ is the unconstrained minimizer of $\mathcal{L}(M)$, we have $\mathcal{L}(M^{(0)}) \leq \mathcal{L}(M^*)$. Since $\mathcal{L}(\cdot)$ is strongly convex, we have

$$0 \geq \mathcal{L}(M^{(0)}) - \mathcal{L}(M^*) \geq \langle \nabla \mathcal{L}(M^*), M^{(0)} - M^* \rangle + \frac{\kappa_\mu}{2} \|M^{(0)} - M^*\|_F^2.$$

We then have

$$\|M^{(0)} - M^*\|_F^2 \leq -\frac{2}{\kappa_\mu} \langle \nabla \mathcal{L}(M^*), M^{(0)} - M^* \rangle \leq \frac{2}{\kappa_\mu} \|\nabla \mathcal{L}(M^*)\|_F \cdot \|M^{(0)} - M^*\|_F,$$

and hence

$$\|M^{(0)} - M^*\|_F \leq \frac{2}{\kappa_\mu} \|\nabla \mathcal{L}(M^*)\|_F \leq \frac{2\sqrt{p}\lambda}{\kappa_\mu}.$$

For large enough $m$, this error bound can be small and Lemma 2 in [28] gives

$$d^2\left(V^{(0)}, V^*\right) \leq \frac{2}{\sqrt{2}-1} \cdot \frac{\|M^{(0)} - M^*\|_F}{\sigma_r(M^*)} \leq \frac{20p\lambda^2}{\kappa_\mu^2 \cdot \sigma_r(M^*)}. \tag{A.12}$$

$\square$

## A.4 Proof of Theorem 4.5.

*Proof.* According to Lemma 4.3 and Lemma 4.4, the initialization $M^{(0)}$ satisfies $\|M^{(0)} - M^*\|_F \leq C$ as long as $m \geq 4C_0 p^2/\kappa_\mu^2$. Furthermore, Lemma 4.1 shows that the objective function $\mathcal{L}(\cdot)$ is strongly convex and smooth. Therefore we apply Lemma 3 in [28] and obtain

$$d^2\left(V^{(t+1)}, V^*\right) \leq \left(1 - \eta \cdot \frac{2}{5}\mu_{\min}\sigma_M\right) \cdot d^2\left(V^{(t)}, V^*\right) + \eta \cdot \frac{\kappa_L + \kappa_\mu}{\kappa_L \cdot \kappa_\mu} \cdot e_{\text{stat}}^2, \tag{A.13}$$

where $\mu_{\min} = \frac{1}{8}\frac{\kappa_\mu \kappa_L}{\kappa_\mu + \kappa_L}$ and $\sigma_M = \|M^*\|_2$. Define the contraction value

$$\beta = 1 - \eta \cdot \frac{2}{5}\mu_{\min}\sigma_M < 1, \tag{A.14}$$

we can iteratively apply (A.13) for each $t = 1, 2, ..., T$ and obtain

$$d^2\left(V^{(T)}, V^*\right) \leq \beta^T d^2\left(V^{(0)}, V^*\right) + \frac{\eta}{1-\beta} \cdot \frac{\kappa_L + \kappa_\mu}{\kappa_L \cdot \kappa_\mu} \cdot e_{\text{stat}}^2, \tag{A.15}$$

which shows linear convergence up to statistical error. For large enough $T$, the final error is given by

$$\begin{aligned}
\frac{\eta}{1-\beta} \cdot \frac{\kappa_L + \kappa_\mu}{\kappa_L \cdot \kappa_\mu} \cdot e_{\text{stat}}^2 &= \frac{5}{2\mu_{\min}\sigma_M} \cdot \frac{\kappa_L + \kappa_\mu}{\kappa_L \cdot \kappa_\mu} \cdot e_{\text{stat}}^2 \\
&= \frac{20}{\sigma_M} \cdot \left(\frac{\kappa_L + \kappa_\mu}{\kappa_L \cdot \kappa_\mu}\right)^2 \cdot e_{\text{stat}}^2 \\
&\leq \frac{80}{\sigma_M} \cdot \frac{e_{\text{stat}}^2}{\kappa_\mu^2}.
\end{aligned} \tag{A.16}$$

Together with (4.6) we see that this gives exactly the same rate as the convex relaxation method (4.3).

$\square$