[Reviews · NeurIPS 2018]

Reviewer 1



Summary: The paper analyzes Gaussian embedding in the exponential-family embedding framework of Rudolph et al NIPS'16. The primary contribution of the paper is to obtain convergence guarantees for two estimation procedures considered in the paper for Gaussian embeddings. The context of each node induces a graph structure over the node and the authors (somewhat) relate convergence with properties of the graph (e.g. the spectral norm of the adjacency matrix). Clarity and Originality: The paper is clearly written and easy to follow. A contribution of the paper is to view the exponential-family embedding model as a graphical model. I have the following questions: 1) How are repeated observations handled? For instance in the word embedding case how are repeated words handled? It seems like each node of the graph corresponds to a unique word? If this is the case, then the flexibility of the exponential-family embeddings model is greatly reduced. 2) The joint distribution over the nodes is not a graphical model: (i) its not a directed graphical model since there can be loops in the graph (ii) its not an undirected graphical model becuase the joint distribution is not a Gibbs distribution i.e. doesn't factor over maximal cliques. 3) How do you leverage the results from the literature of graphical models for estimation? 4) The estimation procedure is essentially the loss function from Rudolph et al specialized for the Gaussian case. However, it is strange to ignore the unknown covariances matrix during estimation --- this is reasonable if the Gaussians are isotropic which does not appear to be the case here. Minor comments: In Lemma 4.1 additional constants \kappa_mu and \kappa_L don't need to be introduced. (4.2) can be directly stated in terms of c_min and c_max. Originality and Significance: The primary contribution of the paper is the theoretical analysis of Gaussian embeddings. The paper makes a number of simplifying assumptions in order to facilitate theoretical analyses. However the assumption that the embedding matrices (V_i's) are the same is problematic. This greatly reduces the flexibility of exponential-family embeddings model. Experimental evidence needs to be provided for the usefulness of the simplified model. I am also not convinced if Assumption SC holds in any reasonable setting. Its very unlikely that the spectral norm of the \Sigma_{col}^{1/2}, which is a pm x pm matrix, is constant --- the spectral norm grows with both the number of variables m and the dictionary size p. Also, where does this constant (\rho) show up in the bound in Theorem 4.2? This is arguably the weakest part of the paper. While the analysis is promising, it is preliminary and the authors should further exploit different properties of the graph to provide bounds for the spectral norm of A and \Sigma_{col}. The authors should additionally provide empirical evidence through simulation experiments on the effect of the covariance matrices on estimation. === Post rebuttal === I think the authors for clarifying some of the issues in the rebuttal. Here is my final assessment of the paper after going through the rebuttal, the paper, other reviews and some related work. The primary novelty of the paper is to view the exponential family embedding model as an instance of the mixed graphical model framework of Chen et al (2014). This allows them to show that the conditional distributions in the Gaussian embedding model induce a valid joint distribution, parameterized by a covariance matrix \Sigma_{col}, over the data (x_j). Due to this the authors are able to obtain theoretical guarantees for two estimation procedures considered in the paper by imposing suitable assumptions on the matrix \Sigma_{col}. The other contribution of the paper is to develop examples to motivate and justify their assumptions. On second reading I think the examples are reasonable --- the identity covariance matrices used in the examples is a common simplification used in the literature for exponential family embeddings. However, both the convex and non-convex method and their analysis is pretty standard. The analysis of the convex method is an invocation of the restricted strong convexity framework of Negahban and Wainwright and all the authors need to show is that the quadratic objective is strongly convex --- which is straight forward. The analysis of the non-convex method is again a straight-forward invocation of the results of Yu et al (2018). The analysis for the Gaussian case would be complete if the authors do the following: 1) Keep the link function in the analysis and ideally state their results with respect to general link functions, 2) keep the embedding and context matrix separate --- the matrix V_j and V_k respectively in Eq 2.2.This would make the results relevant and faithful to the Gaussian embedding case introduced in Rudolph et al.

Reviewer 2



The primary contributions of this paper seem to be applying the convex relaxation technique and nonconvex low rank factorization approach to the exponential family model. The earlier sections appear to be review and background. Section 4 appears to be mostly standard applications of existing theory to this specific problem. I don't think it is quite novel enough. Although the authors start out with discussion general exponential family embeddings, their theorems only apply to the Gaussian setting. I would probably remove the Poisson example, since it's not analyzed. I think this paper is borderline for NIPS.

Reviewer 3



Review of "Provable Gaussian Embedding with One Observation" This paper considers the problem of learning an exponential family embedding (i.e., the parameters of the model) defined on a graph. The (conditional) model describes how a measurement at each node is generated based on (i.e., given) the measurements at neighboring nodes. The neighborhood is determined based on the graph. The authors focus on the Gaussian embedding model and propose a conditional log-likelihood based inference. The authors propose two procedures: convex relaxation and a non-convex optimization procedures to determine the estimates of model parameters (low rank matrix M). The authors proceed with theoretical results on the convergence rates for both approaches as well as conditions (assumption EC and SC) to achieve these convergence rates. Overall, I think that the paper has all the ingredients, including an important problem, a well define solution, and a thorough theoretical analysis. Minor comments: * Lemma 4.1 - "are hold" => "hold" * Lemma 4.1 - 'absolute constants' => 'universal constants?'